# A Physicochemical Stability Study of Pembrolizumab Vial Leftovers: Let Us Stop Pouring Good Money Down the Drain

**DOI:** 10.3390/pharmacy13010022

**Published:** 2025-02-08

**Authors:** Alexandra Porlier, Pierre-Yves Gagnon, Valérie Chénard, Marc Veillette, Nicolas Bertrand, Caroline Duchaine, Chantale Simard, Benoît Drolet

**Affiliations:** 1Research Centre, Quebec Heart & Lung Institute, Laval University, Quebec City, QC G1V 4G5, Canada; alexandra.porlier@criucpq.ulaval.ca (A.P.); marc.veillette@criucpq.ulaval.ca (M.V.); caroline.duchaine@bcm.ulaval.ca (C.D.); chantale.simard@pha.ulaval.ca (C.S.); 2Department of Pharmacy, Quebec Heart & Lung Institute, Laval University, Quebec City, QC G1V 0A6, Canada; pierre-yves.gagnon@ssss.gouv.qc.ca; 3Research Centre, CHU de Québec, Laval University, Quebec City, QC G1V 0A6, Canada; valerie.chenard@crchudequebec.ulaval.ca (V.C.); nicolas.bertrand@pha.ulaval.ca (N.B.); 4Faculty of Pharmacy, Laval University, Quebec City, QC G1V 0A6, Canada; 5Faculty of Sciences and Engineering, Department of Biochemistry, Microbiology and Bioinformatics, Laval University, Quebec City, QC G1V 0A6, Canada

**Keywords:** pembrolizumab, vial leftovers, stability

## Abstract

Background: Pembrolizumab is a monoclonal antibody (mAb) approved for treating Non-Small Cell Lung Cancer (NSCLC), melanoma and lymphomas. Commercialized in single-size (100 mg/4 mL) vials, the pembrolizumab solution contains no preservative. As such, the manufacturer recommends using pembrolizumab vials only once, and thus, to rapidly dispose of any unused portion. Thus, appreciable amounts of this costly product are wasted. Objective: To evaluate the physical, chemical and microbiological stability of pembrolizumab vial leftovers stored at room temperature or at 4 °C, 7 and 14 days after first vial puncturing. Methods: Following pH assessments, submicronic aggregation and turbidity of pembrolizumab were measured by dynamic light scattering (DLS) and spectrophotometry, respectively. In addition, SE-HPLC (size-exclusion high-performance liquid chromatography), IEX-HPLC (ion exchange HPLC) and peptide mapping HPLC served to respectively evaluate aggregation and fragmentation, distribution of charge and primary structure of pembrolizumab. Incubation at 37 °C for 48 h of pembrolizumab vial leftovers on blood agar plates was used to determine their microbiological stability. Results: Physical, chemical and microbiological stability of pembrolizumab leftovers was demonstrated for at least two full weeks. Conclusions: These results argue forcefully in favor of allowing prolongation of pembrolizumab vial leftovers usage well beyond a single day.

## 1. Introduction

Pembrolizumab (Keytruda^®^) is an IgG4/κ isotype monoclonal antibody (mAb) designed to sit on the PD-1 (programmed cell death 1) receptor. In doing so, it prevents both its ligands, PD-L1 and PD-L2, from interacting with the receptor [1]. As a key immune checkpoint, the PD-1 pathway may be stimulated by cells in the immediate vicinity of the tumor to escape activated T-lymphocyte immune control. Pembrolizumab, by blocking this pathway, promotes efficient T-lymphocyte function, thus favoring tumor regression [2]. In the United States and Canada, pembrolizumab is currently approved for treating many subtypes of cancers, including melanoma, Non-Small Cell Lung Cancer (NSCLC), classical Hodgkin and primary mediastinal large ß-cell lymphomas; head and neck, esophageal, gastric, urothelial, cervical, biliary tract and bladder cancers; hepatocellular, renal, endometrial and cutaneous squamous cell carcinomas; as well as Triple-Negative Breast Cancer (TNBc) [3,4]. In advanced NSCLC patients, when compared to docetaxel or platinum-based chemotherapy, pembrolizumab was shown to increase both overall and progression-free survival [5,6]. Moreover, when compared to a number of chemotherapeutic agents, such as paclitaxel and carboplatin, pembrolizumab performed better in safety, overall survival and progression-free survival [7]. In addition, when used for melanoma and compared to ipilimumab-based treatment, pembrolizumab was shown to increase progression-free survival [8].

Pembrolizumab is marketed in single-use vials containing no preservatives, as an undiluted solution (100 mg/4 mL) to be further diluted in saline before intravenous infusion. Such an injectable product containing no preservatives is, thus, considered as ‘High-risk’ by the USP (United States Pharmacopeia) [9], essentially as a microbiological precaution. Indeed, the USP considers that 24 h at room temperature is the maximum microbiological BUD (Beyond-use Dating) for ‘High-risk’ products, such as most monoclonal antibodies (mAbs). Consequently, the manufacturer (Merck Sharpe & Dohme, Rahway, NJ, USA) recommends discarding any unused portion left in the vial and administering pembrolizumab solution quickly after it is prepared. In the event the solution is not infused immediately, it must be discarded after 6 h at room temperature or after 96 h under refrigeration [3,4].

Although the therapeutic efficacy of pembrolizumab has been shown, it comes at a price, which is very high. In Canada (including the province of Québec [10]) the acquisition cost is CAD 4400.00 (USD 3212.00) per 100 mg vial (100 mg/4 mL) or CAD 1100.00/mL [11]. The pan-Canadian Oncology Drug Review (pCODR) has determined that treating a single patient on a 2 mg/kg every 3 weeks dosing regimen can cost as much as CAD 140,029.00 per year or CAD 8237.00 per 28-day course of treatment [11]. A pembrolizumab dose of 2 mg/kg was initially approved in 2014 by the FDA and in 2015 by Health Canada [3,4]. As the years went by, a flat dose of 200 mg was also approved in both countries for all pembrolizumab’s indications [3,4]. At first, the flat dosing option was well received by clinicians, as being much more convenient, while suggesting the end of product wastage. However, it was soon realized that this ‘one size fits all’ practice was exposing most patients to higher than needed doses, considering that the better efficacy of such doses has never been proven [11]. Thus, when thinking about it, flat dosing is for most patients a form of wastage that still inflates the final treatment costs. It was, therefore, established that using pembrolizumab 2 mg/kg is a fair clinical practice that turns out to be more cost-effective while making no compromise on efficacy [11].

In fact, the cost of pembrolizumab wastage goes beyond imagination, reaching hundreds of millions of dollars yearly worldwide [12]. Of note, Canadian provincial healthcare systems are not escaping from this tsunami of expenses. A retrospective analysis of melanoma and NSCLC patients in all six British Columbia Cancer Regional Centres performed for fiscal years 2017 and 2018 [11] showed the amplitude of the problem. In this study, a total of 2948 pembrolizumab doses were administered for treating 202 patients with NSCLC and 182 others with melanoma. Without any vial sharing (using vial leftovers to treat other patients), the amount of drug wasted by these six BC Cancer Treatment Centres while treating those 384 patients was measured at CAD 6,682,460.40. With optimal (and theoretical!) vial sharing, the amount of drug saved was estimated at CAD 3,207,600.00, which is appreciable but still leaving a whopping CAD 3,474,860.40 in losses, and still representing 15.25% wastage of the total product despite optimal 100 mg vial sharing assumptions.

Apart from drug dosing, other factors increase the costs of mAbs such as pembrolizumab, including single-use vials and same-day administration practice, cancellation from patients for any given reason, and even no-shows [12,13,14]. However, the current ‘single-use vials’ and ‘same-day administration’ practice is, by far, the leading contributing factor to the overall wastage of mAbs [15]. For instance, in a large cancer treatment center such as our own, the high number of patients treated with pembrolizumab for NSCLC, along with tight drug administration scheduling, allow efficient vial sharing and minimize drug wastage. Yet, despite all these vial usage-optimizing efforts, our pharmacists have evaluated that pembrolizumab wastage reached a total of 92 mL for the fiscal year 2022–2023, which is equivalent to 23 full vials of undiluted pembrolizumab (Keytruda^®^ 100 mg/4 mL). With an acquisition cost of CAD 4400.00 per vial, this represents at least CAD 101,200.00 yearly, or a CAD 8433.33 average monthly loss for our institution only, on this single product! (personal communication).

Interestingly, most of the concerns raised by mAb manufacturers to justify the single-use of vials and rapid discarding of leftovers are based on potential threats of a microbiological nature [3,4]. Yet, a number of studies have demonstrated that numerous extensively used, preservative-free mAbs remain highly stable even after dilution in saline or as undiluted vial leftovers way over 24 h [15,16,17,18,19], provided that they were handled under appropriate aseptic settings, which are required and tightly controlled in any cancer treatment center. In addition, in any cancer treatment center. Thus, once aseptic conditions are applied, the overall stability of mAbs mainly depends on their intrinsic physicochemical properties.

Although it has been marketed in the US since 2014 and in Canada since 2015, data on pembrolizumab’s physicochemical stability, specifically in the form of undiluted vial leftovers, are still very scarce [16]. Interestingly, a recent study by Arnamo et al. [20] showed that by using size exclusion chromatography (SE-HPLC) and dynamic light scattering (DLS), pembrolizumab vial leftovers are stable from a physicochemical standpoint for at least two weeks when kept refrigerated. They also showed the biological stability of these vial leftovers by use of an Enzyme-Linked Immuno-Sorbent Assay (ELISA). However, this study provided no data on microbiological stability whatsoever. The present study was, therefore, aimed at complementing the study of Arnamo et al. [20] by evaluating the physicochemical stability of pembrolizumab vial leftovers using further sets of new experiments using other additional assays and by providing the first evidence suggesting microbiological stability of these leftovers when stored at either room temperature or 4 °C on Days 0, 7 and 14 after first puncturing.

## 2. Methods

### 2.1. Pembrolizumab Samples

Unopened Keytruda^®^ vials (pembrolizumab 100 mg/4 mL, Merck Canada Inc., Kirkland, QC, Canada), lot numbers A102541, A102193 and X012124, were obtained from the Pharmacy Unit of our hospital. A total of 3 punctures (only once on Day 0, 7 and 14) were performed aseptically in each vial with an 18 G × 1” needle syringe.

### 2.2. Visual Inspections

In a sterile hood, before handling vials on Days 0, 7 and 14, inspection was performed to look for suspended particles, turbidity, signs of formation of aggregates or gas or any colour change, as described previously. This inspection was always performed by the same person using the naked eye without background. The vials were protected from sunlight at all times, as described previously [21].

### 2.3. Turbidity

A UV-VIS 1800 spectrophotometer (Shimadzu, Columbia, MD, USA) was used to measure the turbidity of pembrolizumab solution samples in triplicates at room temperature. Ultraviolet (UV) absorbance of samples was obtained at two wavelengths, 280 nm (*A*_*λ*280_) and 350 nm (*A*_*λ*350_), and used to calculate the aggregation index (AI) using this formula, as described previously [21].$$\mathrm{AI}=\frac{A\lambda 350}{A\lambda 280-A\lambda 350}\times 100$$

When AI is under 10, it means there are no soluble aggregates. Pembrolizumab 100 mg/4 mL vial leftovers were diluted in demineralized water (filtered at 0.22 µm) at 1 mg/mL (final concentration) to take into account the detection capacity of the apparatus.

### 2.4. Dynamic Light Scattering (DLS) to Assess Aggregation of Submicronic Particles

At different time intervals, pembrolizumab’s hydrodynamic diameter was compared by dynamic light scattering (DLS), as described previously [21]. DLS serves to detect aggregates of 1 nm–10 µm size. Samples of pembrolizumab vial leftover solution (100 µL) were tested at 25 °C in triplicates on a Zetasizer apparatus (Malvern, UK). The Z-average calculated by the instrument is defined as the intensity-weighted averaged hydrodynamic diameter. The polydispersity index (PDI) was also evaluated. When representative population of pembrolizumab particles had no deviation superior to 1 nm from their normal mean hydrodynamic diameter, the solution was considered stable and monodispersed (PDI smaller than 0.1 means less than 10% change in mean normal diameter and no aggregation).

### 2.5. pH Measurements

A pH meter was used to measure the pH of pembrolizumab undiluted vial leftovers in triplicates, as described previously [21]. The Day 0 value was used to compare the pH of pembrolizumab leftovers at Day 7 and Day 14. A variation smaller than half a unit (0.5) of pH from Day 0 value was considered to show stability of the pH of the solution.

### 2.6. Size-Exclusion High-Performance Liquid Chromatography (SE-HPLC) to Assess the Aggregation of Pembrolizumab

SE-HPLC was used to determine (in triplicates) the aggregation of pembrolizumab in solution, as described previously [21]. An SIL-20ACHT automatic sample injector, an SPD-20A UV detector and an LC-20AT pump (Shimadzu, Columbia, MD, USA) were used at 25 °C. An isocratic method was performed at 0.6 mL/min using a buffer solution A of 0.2 M potassium phosphate buffer with 0.25 M potassium chloride, pH 6.2 [22]; aXBridge Premier Protein SEC 250 Å column with its guard column was used (Waters, Mississauga, ON, Canada). Each pembrolizumab solution sample was diluted to 1 mg/mL in the buffer solution A and filtered. Thereafter, 25 µL of pembrolizumab (1 mg/mL) was injected into the system. Pembrolizumab’s peak areas were obtained at a wavelength of 280 nm using UV absorbance, and Day 0 data were considered as reference values. The EZ Start software package version 7.4 from Shimadzu was used to collect and analyze the chromatograms. USP <129> monograph criteria [22] were applied to assess pembrolizumab stability.

### 2.7. Ion Exchange HPLC (IEX-HPLC) to Assess Pembrolizumab’s Charge Distribution

Deamidation-induced variations in charge distribution of pembrolizumab’s main, acidic and basic species were assessed by IEX-HPLC, according to criteria of USP <129> monograph [22].

IEX-HPLC was used to perform analyses in triplicates at 30 °C in a gradient mode, as described previously [21]. Two mobile phases (A and B) were composed of 0.02 M MES (4-morpholineethanesulfonic acid, Millipore Sigma, St. Louis, MO, USA), pH 6.2, and 0.5 M of NaCl was added to the mobile phase B. BioResolve SCX mAb column and SCX mAb VanGuard FIT guard column (both from Waters, Milford, MA, USA) were used to perform this gradient method at a flow rate of 0.4 mL/min. Each pembrolizumab solution sample was diluted (1 mg/mL) by use of the mobile phase A, filtered before the transfer into the HPLC vial, and then injected (50 microliters) into the system. The gradient conditions used are summarized as follows:

      **Time**      

      **%B**      
Initial0%60%126%3612%5118%52100%530%680%

Pembrolizumab’s peak areas were obtained at a wavelength of 280 nm by UV absorbance, and Day 0 data were considered as reference values [23].

### 2.8. Peptide Mapping HPLC to Determine the Primary Structure of Pembrolizumab

Peptide mapping HPLC was used to determine the primary structure of pembrolizumab, as described previously [21]. Samples of pembrolizumab solution (100 mg/4 mL) were diluted to 1 mg/mL with a digestion buffer made of 50 mM NH_4_HCO_3_, pH 7.8 (Millipore Sigma). Fifty microliters (50 µL) of this diluted solution (in duplicates) were further diluted in an extra volume (10 µL) of the digestion buffer [23]. To enhance the enzymatic digestion, a solution of 0.1% Rapigest^®^ reagent (Waters) was added to the diluted samples. The diluted samples were heated at 80 °C for 20 min to denature the antibody (pembrolizumab). After this step, the denatured samples were cooled down on a bench for a few minutes [23]. The reduction step consisted of adding 1 µL of 0.22 M dl-dithiothreitol (DTT, Millipore Sigma) to all denatured samples, which were then mixed and warmed at 37 °C for 60 min. All reduced samples were then mixed with 1 µL of 0.66 M iodoacetamide (IAA, Millipore Sigma) incubated in darkness for 30 min at RT [23]. The alkylated samples were then digested in 1.5 µg of trypsin (Promega, Madison, WI, USA). The digested samples were then agitated and warmed at 37 °C overnight (for 18 h) [23]. The enzymatic reaction was stopped with a solution of 25% trifluoroacetic acid (TFA, Millipore Sigma). The preparation was then agitated and warmed at 37 °C for 30 min [23]. After this incubation, the digested pembrolizumab samples were centrifugated at 13,000 rpm for 10 min at RT, and approximately 50 µL of the supernatants were used for HPLC analysis [23].

### 2.9. HPLC Analyses

These analyses were performed, as described previously [21], at 40 °C. Two mobile phases (A and B) were prepared by adding trifluoroacetic acid (0.1%) in water (A) or (B) in acetonitrile (Fisher Scientific, Whitby, ON, Canada) to be used later in gradient mode. The specific conditions of the gradient were as follows:

      **Time**      

      **%B**      
Initial0.5%20.5%6250%6595%6695%800.5%

An XSelect Premier CSH C18 130 Å column with its guard column (both from Waters) was used for this gradient method at a flow rate of 0.5 mL/min. Samples as triplicates of 25 µL each were injected into the system. Peaks of pembrolizumab were obtained by UV absorbance detection at two wavelengths: 214 and 280 nm. The Day 0 chromatograms of pembrolizumab peptide mapping served as baseline and reference values. The USP <1055> monograph [24] criteria were used to evaluate the stability of pembrolizumab.

### 2.10. Microbiological Stability Assessment

Pembrolizumab vial leftovers’ microbiological stability was evaluated, as described previously [21,25,26]. Briefly, all pembrolizumab (100 mg/4 mL) solution samples (100 µL) were aseptically withdrawn from pembrolizumab vials and inoculated on plates containing 5% blood agar, as triplicates. These plates were placed in an incubator for 48 h at 37 °C before detection of colonies and counting. All this work was conducted while fully complying with the ISO 14644-01 norm in effect at the Pharmacy Department of IUCPQ-UL [26].

### 2.11. Statistical Analyses

The GraphPad Prism software, version 10.2.2 (La Jolla, CA, USA), was used to perform statistical analyses. Two-way repeated measures ANOVA or Student’s paired *t*-test were used to compare data. All parameters were presented as mean ± SD, and *p* values < 0.05 were considered statistically significant.

## 3. Results

### 3.1. Physicochemical Stability Assessment

No evidence of colour changes, gas formation, turbidity, suspended particles or formation of large aggregates were ever noticed after visual inspection. Table 1 and Table 2 show the data acquired from pembrolizumab vial leftovers on Days zero, seven and fourteen, when they were stored at either room temperature or in a refrigerator at 4 °C. All data were compared as Day zero vs. Day seven or as Day zero vs. Day fourteen.

There was no difference in turbidity after 7 or 14 days at both storage temperatures (room or 4 °C). As seen in Table 1 and Table 2, the aggregation index (AI) stayed highly stable and far below 10 at all times, ruling out the presence of visible or subvisible soluble aggregates at any given time.

With a variation of less than 1 nm at any given time, the mean hydrodynamic diameter of pembrolizumab stayed well within acceptance limits at either room temperature or at 4 °C. The polydispersity index (PDI) has never reached 0.1 at any moment at either room temperature or at 4 °C. Pembrolizumab vial content was considered as a monodispersion, implying the absence of submicronic aggregation, further demonstrating stability of the solution throughout all the experiments at both room temperature and 4 °C.

With a deviation of less than 0.5 unit at all times, the pH of undiluted pembrolizumab vials remained stable at both room temperature and 4 °C. Indeed, all values remained within the acceptable pH range (5.2–5.8) throughout all experiments.

SE-HPLC experiments with pembrolizumab vial leftovers revealed four peaks: the first representing the oligomers of pembrolizumab, the two main peaks illustrating the well-characterized monomeric and dimeric forms of pembrolizumab [1], and a fourth peak of pembrolizumab fragments (a seen in Figure 1 and its inset). These peaks were observed in all samples. At all times and at either room temperature or 4 °C, the AUC (area under the curve) of the monomeric and dimeric peaks of pembrolizumab accounted for nearly all (>99%) of the AUC of all peaks put together (Table 1 and Table 2). As a result, this left a very small portion of the total AUC representing the two remaining peaks of oligomers and fragments of pembrolizumab. These two combined have never reached 0.5% of the total AUC at either room temperature or 4 °C, showing that pembrolizumab did not aggregate or fragment, thus, again, demonstrating high stability.

IEX-HPLC chromatograms displayed pembrolizumab’s distinctive peaks: the principal species, a number of pembrolizumab acid species and a smaller set of pembrolizumab basic species (as seen in black as upward peaks of Figure 2A). All these pembrolizumab species (principal, acid and basic) remained constant in terms of relative proportions at any given time and at either room temperature or 4 °C (Table 1 and Table 2 and downward peaks (in red) of Figure 2A + panels B and C of Figure 2).

Chromatograms generated in peptide mapping experiments (Figure 3A,B) revealed that at two different wavelengths (214 and 280 nm), pembrolizumab’s primary structure was not affected at all when vial leftovers were stored for as long as 2 weeks at either room temperature or at 4 °C. Indeed, all pembrolizumab chromatograms obtained at either 214 or 280 nm are nearly superimposable.

### 3.2. Microbiological Stability Assessment

All pembrolizumab leftover samples, withdrawn from the vial on Day 0, 7 or 14 and incubated in triplicates at 37 °C during 48 h on blood agar plates tested negative for any bacterial or fungal growth.

## 4. Discussion

Pembrolizumab vial leftovers’ physicochemical stability, as a function of time and storage temperature, was thoroughly tested. When stored at either room temperature or 4 °C and handled aseptically, as recommended, pembrolizumab vial leftovers were shown to remain highly stable from a physicochemical standpoint for a minimum of 14 days. Moreover, the microbiological assay performed on these pembrolizumab vial leftovers showed no bacterial or fungal growth, suggesting microbiological stability as well.

Neither turbidimetry, DLS nor SE-HPLC showed physical instability. No pembrolizumab corpuscle with a larger hydrodynamic diameter was ever detected, ruling out nucleation, a phenomenon generally occurring before aggregation [15]. In addition, SE-HPLC experiments have shown that the relative percentages of the main peaks (monomeric and dimeric pembrolizumab) or of the remaining peaks (representing pembrolizumab oligomers and fragments) hardly ever changed at all, thus demonstrating that aggregation or fragmentation of pembrolizumab has not happened, even after two full weeks of storage at either room temperature or 4 °C. Moreover, the relative percentages of pembrolizumab main component, as well as acidic and basic variants, remained virtually unchanged over time, as shown by IEX-HPLC experiments, thus, again, confirming the remarkable stability of pembrolizumab vial leftovers up to two full weeks when stored at either room temperature or 4 °C.

Potential protein degradation and site-specific assessment of chemical reactions, such as oxidation or deamidation of pembrolizumab, was performed by peptide mapping HPLC. Pembrolizumab’s primary structure was not affected by 14 days of storage at either room temperature or 4 °C. This did not come as a surprise, as changes in the proportion of pembrolizumab’s main variant were way too small (<1.6%, as measured by IEX-HPLC [15]) to detect any noticeable modification of the primary structure.

Pembrolizumab’s microbiological stability was also suggested in all vial leftover samples tested. Indeed, whether the samples were from vial leftovers stored at RT or in a refrigerator at 4 °C for 7 or 14 days, no bacterial or fungal growth (including yeasts) of any kind was ever observed in any blood agar culture plates when they were incubated for 48 h at 37 °C.

This is in perfect agreement with a similar study with durvalumab [21] and with a recent study [16] showing that from a microbiological standpoint, it is acceptable to extend storage and to further use monoclonal antibody leftovers, as the overall risk of contamination is very low (0.05%), even in multi-punctured vials. As an additional convincing example, Das et al. [18] have not seen any infection or inflammation in the eyes of their 221 consecutive patients when receiving a total of 973 intravitreal injections of bevacizumab, even without prior aliquoting of vial leftovers, and thus, after hundreds of direct withdrawals from these vials. In fact, their vial leftovers were shown to remain sterile for at least a week when kept refrigerated.

Numerous studies from around the world have shown that mAbs, provided that they are kept in appropriate conditions and adequately handled by qualified people, should be considered far more stable than previously thought and specified in the official monographs of their respective manufacturers [16,17,18,19,27,28,29,30,31]. Indeed, it has even been shown that mAbs have to be exposed to high temperatures for an extended period of time to observe relevant physicochemical alterations [32]. The present report is in agreement with all these studies. In fact, we demonstrated physicochemical stability of undiluted (100 mg/4 mL) pembrolizumab vial leftovers for at least two weeks when conserved at either room temperature or 4 °C. Our microbiological data suggest stability as well over the same period and in the same storage conditions. The current study also corroborates similar results concerning pembrolizumab’s physicochemical stability as an admixture solution in saline IV infusion bags [20,31] or as vial leftovers, such as recently described by Arnamo et al. [20]. Yet, of utmost importance, the present study is the first to suggest that pembrolizumab vial leftovers might be stable for at least two weeks, from a microbiological standpoint, when stored at either room temperature (RT) or at 4 °C. The present study also reinforces Arnamo’s physicochemical stability data by adding IEX-HPLC data confirming the stability of pembrolizumab and both its acidic and basic variants, as well as demonstrating the stability of pembrolizumab’s primary structure by peptide mapping HPLC. A possible limitation of the present study is that we did not assess the biological stability of pembrolizumab vial leftovers. However, by using flow cytometry and ELISA testing, pembrolizumab’s biological stability after dilution and storage in saline bags at either 1 or 4 mg/mL was shown by Acramel et al. [33] to reach one full week at room temperature and four full weeks at 4 °C, which was also recently corroborated by Arnamo et al. [20]. Another potential limitation of this study is the potential instability risks that might come from the progressive increase in interfacial contact of the pembrolizumab solution with glass over time (less solution in the vial relative to the same contact surface).

Indeed, glass delamination is a phenomenon known to happen with injectable drug solutions stored for several months, including some biologics such as antibodies. This can lead to the generation of both visible and subvisible particles. Delamination results from the chemical attack on the glass surface. In fact, glass attack, in this case, mostly results from ion exchange or dissolution. Ion exchange happens when water manages to diffuse into the glass, leading to the exchange of H+ with alkali ions. On the other hand, dissolution is mainly caused by hydroxide ions (OH-) attacking the glass silicate backbone. Overall, these two mechanisms could lead to the formation of a leached layer, potentially detaching from the glass surface.

Yet, the data presented in the present paper do not suggest that increased interfacial contact with glass is a relevant issue with pembrolizumab vial leftovers. Indeed, the assays we used to evaluate the progressive formation of visible and subvisible particles (visual inspection and spectrophotometry) did not show any relevant formation of such particles. More importantly, our ion exchange chromatography (IEX-HPLC) data showed that the relative proportion of pembrolizumab and both its acidic and basic variants did not change over a period of 14 days at either 4 °C or RT. Glass leaching due to ion exchange is, thus, not considered to be of any relevance in the case of pembrolizumab vial leftovers.

## 5. Conclusions

The long-standing single-use vial and same-day administration clinical practice associated with pembrolizumab should be revised. Indeed, these new sets of data show that pembrolizumab vial leftovers could be safely administered at least 14 days after the first product withdrawal from the vial. This would, therefore, offer the possibility of reallocating vial leftovers to other patients within the next few days instead of discarding them on a daily basis. This would further optimize the usage of the product and reduce wastage to nearly zero. Allowing later usage of pembrolizumab vial leftovers that, for any given reason, cannot be infused to other patients within the same day would, therefore, lead to huge savings by eliminating costly wastage.

## Figures and Tables

**Figure 1 pharmacy-13-00022-f001:**
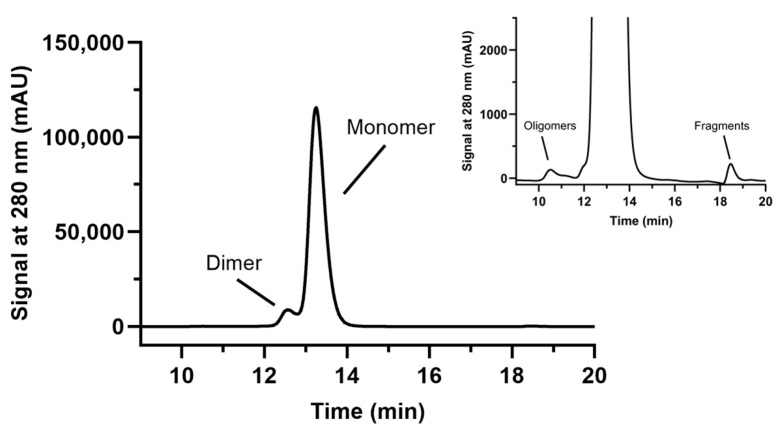
SE-HPLC elution profile of dimeric and monomeric pembrolizumab. Oligomers and fragments are magnified in inset.

**Figure 2 pharmacy-13-00022-f002:**
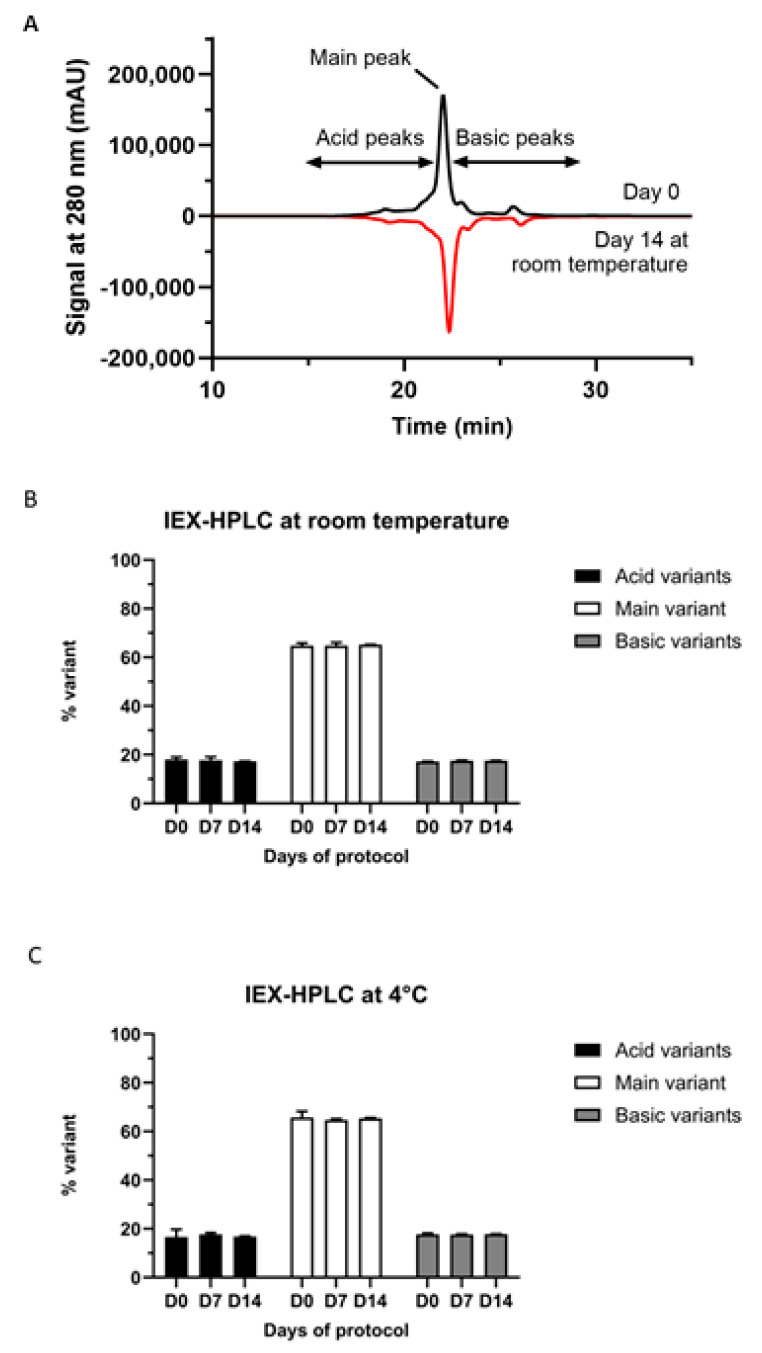
(**A**) IEX-HPLC elution profile of pembrolizumab. A number of peaks of the respective isoforms of pembrolizumab (main, acid and basic) are represented by the black tracing, with Day zero respective proportions. The same data are shown on Day fourteen at RT by the tracing in red. (**B**) Respective percentage of pembrolizumab’s same isoforms at Days zero, seven and fourteen with storage at RT. (**C**) The same data as in panel B but with vials stored at 4 °C.

**Figure 3 pharmacy-13-00022-f003:**
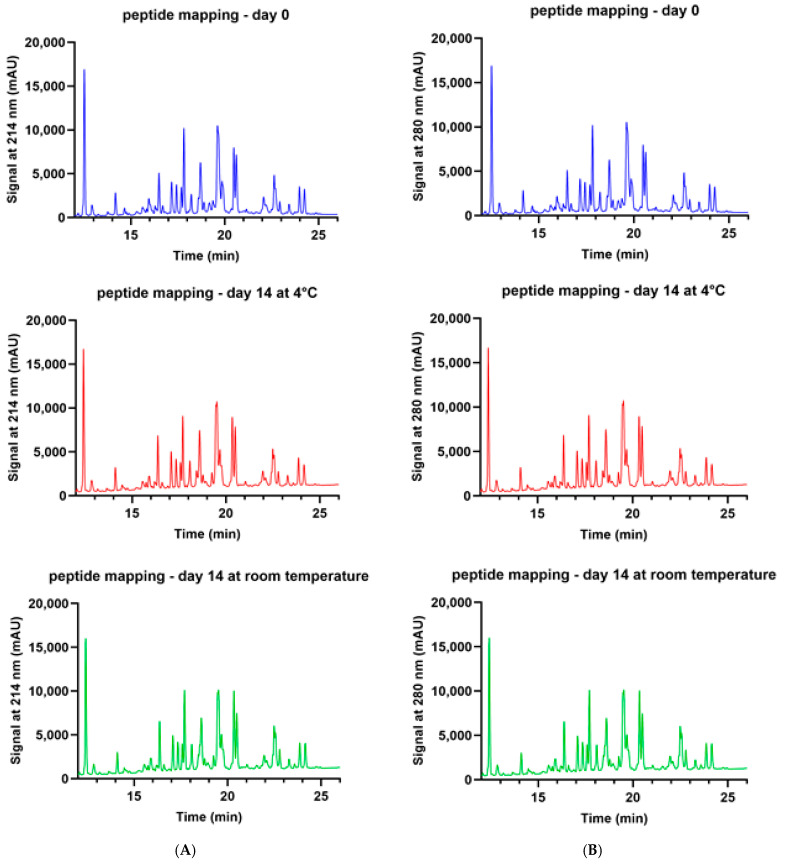
(**A**). Peptide mapping chromatograms obtained by HPLC at 214 nm of pembrolizumab vial leftover solutions on Day zero (upper panel, blue) and on Day fourteen when refrigerated at 4 °C (middle panel, red) or at RT (lower panel, green). (**B**). Peptide mapping chromatograms obtained by HPLC at 280 nm of pembrolizumab vial leftover solutions on Day zero (upper panel, blue) and on Day fourteen when refrigerated at 4 °C (middle panel, red) or at RT (lower panel, green).

**Table 1 pharmacy-13-00022-t001:** Pembrolizumab (100 mg/4 mL) stored at RT (room temperature). All mean ± SD.

Parameters		Day Zero	Day Seven	Day Fourteen	Statistics
Turbidity	Aggregation index (AI %)	0.1288 ± 0.0479	0.1428 ± 0.0195	0.1110 ± 0.0179	Non-significant
Dynamic light scattering (DLS)	Size (nm)	11.31 ± 0.05	11.27 ± 0.06	11.11 ± 0.13	Non-significant
	Polydispersity index (PDI)	0.077 ± 0.004	0.071 ± 0.006	0.070 ± 0.008	Non-significant
pH	Units of pH	5.55 ± 0.01	5.54 ± 0.03	5.54 ± 0.02	Non-significant
SE-HPLC	Variants (%)				
	Oligomeric	0.09 ± 0.02	0.09 ± 0.01	0.09 ± 0.02	Non-significant
	Dimeric	6.29 ± 0.05	6.58 ± 0.02	6.68 ± 0.06	Non-significant
	Monomeric	93.44 ± 0.03	93.14 ± 0.02	93.05 ± 0.05	Non-significant
	Fragmental	0.18 ± 0.01	0.18 ± 0.00	0.18 ± 0.00	Non-significant
IEX-HPLC	Variants (%)				
	Acid peaks	18.0 ± 1.0	17.7 ± 1.4	17.3 ± 0.1	Non-significant
	Main peak	64.8 ± 1.1	64.8 ± 1.3	65.2 ± 0.1	Non-significant
	Basic peaks	17.2 ± 0.1	17.5 ± 0.2	17.5 ± 0.1	Non-significant
Microbiological assay	Colonies	Absent	Absent	Absent	

**Table 2 pharmacy-13-00022-t002:** Pembrolizumab (100 mg/4 mL) refrigerated at 4 °C. All mean ± SD.

Parameters		Day Zero	Day Seven	Day Fourteen	Statistics
Turbidity	Aggregation index (AI %)	0.1346 ± 0.0855	0.1550 ± 0.0138	0.1138 ± 0.0216	Non-significant
Dynamic light scattering (DLS)	Size (nm)	11.21 ± 0.15	11.35 ± 0.05	11.45 ± 0.04	Non-significant
	Polydispersity index (PDI)	0.067 ± 0.013	0.076 ± 0.014	0.085 ± 0.009	Non-significant
pH	Units of pH	5.64 ± 0.13	5.65 ± 0.13	5.67 ± 0.11	Non-significant
SE-HPLC	Variants (%)				
	Oligomeric	0.06 ± 0.02	0.06 ± 0.02	0.08 ± 0.02	Non-significant
	Dimeric	6.08 ± 0.31	6.33 ± 0.27	6.49 ± 0.36	Non-significant
	Monomeric	93.66 ± 0.32	93.43 ± 0.27	93.24 ± 0.36	Non-significant
	Fragmental	0.19 ± 0.00	0.18 ± 0.00	0.19 ± 0.01	Non-significant
IEX-HPLC	Variants (%)				
	Acid peaks	16.6 ± 3.2	17.7 ± 0.7	16.8 ± 0.2	Non-significant
	Main peak	65.7 ± 2.6	64.7 ± 0.5	65.4 ± 0.2	Non-significant
	Basic peaks	17.7 ± 0.5	17.6 ± 0.3	17.8 ± 0.1	Non-significant
Microbiological assay	Colonies	Absent	Absent	Absent	

## Data Availability

The original contributions presented in this study are included in the article. Further inquiries can be directed to the corresponding author(s).

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
