# Peer review of "A Physicochemical Stability Study of Pembrolizumab Vial Leftovers: Let Us Stop Pouring Good Money Down the Drain"

_pharmacy, 2025, doi:10.3390/pharmacy13010022_

Round 1
Reviewer 1 Report
Comments and Suggestions for Authors
The manuscript pharmacy_3431577 entitled “Wastage of Pembrolizumab (Keytruda®) Vial Leftovers. Let’s Stop Pouring Good Money Down the Drain” is a research article that aims to explore the physichochemical stability of pembrolizumab (Keytruda®) solutions as vial leftovers.
In my opinion the subject is timely and of interest. Overall, the paper is quite well written, the scientific rational is well defined and the impact for the scientific and pharmaceutical community quite clear.
I would recommend this manuscript for publication once the following corrections / enhancements / explications are made/given.
IN DETAIL:
Title: The title is attractive but not quite in adequation with the content. The authors performed a physicochemical stability of an mAb (albeit a currently expensive one, I agree) and even if they cleverly make the case and justify their study (see my comments about the introduction section), I still think the title is somewhat misleading, as they did not perform a medico-economic study per se. The authors could keep the essence of their work and part of the original title by changing it (suggestion) to one of the following:
A physicochemical stability of pembrolizumab vial leftovers: Let’s Stop Pouring Good Money Down the Drain
Or
Wastage of Pembrolizumab (Keytruda®) Vial Leftovers: A physicochemical stability
Keywords: I disagree that the term “microbial stability” is adapted, as the authors only evaluated on blood agar plates (which I do not think even qualifies as being a sterility assay, at least not following European ou US Pharmacopoeia recommendations). I would remove this key word.
Abstract: No particular comments.
Introduction:
This section is well written, and the authors clearly put set the rational of their study and the lack of data concerning the solution in leftover vials. However, they do not reference in the introduction the work of Arnamo et al. (ref 39, which was briefly touched upon in the discussion) which also studied the physicochemical stability of leftover for up to 14 days at refrigerated temperature. The authors should clearly indicate how their work position’s itself compared to Arnamo et al. What is lacking in Arnamo et al. that justifies this additional work? Perhaps the authors could also reference what potential instability risks might be present (physical instability due to increased interfacial contact (less solution in the vial, same contact surface)?
Materials and methods:
I really believe that the manuscript needs to be greatly enhanced by providing more data in all the subsections of the materials and methods section. Referencing previously published work is generally considered acceptable if that work is freely available or easily accessed, however this is not the case here. The J Pharm Sci (reference 20) is a journal that publishes manuscripts that are not open access, and technical data sheets from Waters (references 22 and 23) do not possess a DOI and could become unavailable with any notice given by the manufacturer.
The authors must really give adequate and sufficient detail in this manuscript or in the supplementary data file to allow the readers to reproduce the experiments.
In this section the authors should also add a “acceptability criteria” section, describing what modification are or are not considered to be proof of stability / instability.
Results and discussion:
Overall: The results are nicely presented, and the discussion well written. In line with the introduction section, perhaps they could discuss the fact that there was intrinsically a low instability risk (no additional solubilisation, no dilution), only added contact with oxygen and increased interfacial contact with the glass vial (less solution, same glass surface).
Concerning the microbiological test: the authors should tone down a bit their statement about proving the microbiological stability, as firstly I do not think they performed the microbiological assay in line with Sterility Monographs and secondly because this element also depends on manipulator experience and working conditions, and is not guaranteed if the manipulating conditions where to change.
Figure 3 really needs to be bigger, as it is hard to compare chromatograms.
Author Response
"Please see the attachment."

Reviewer 2 Report
Comments and Suggestions for Authors
I read with interests the paper titled "Wastage of pembrolizumab (Keytruda®) vial leftovers. Let’s stop pouring good money down the drain"
I really appreciated the paper as it is, however, I have minor questions that could enhance the manuscript understanding.
1. Why did you use the "artificial" unopened vials to perform the stability tests. This is something that could be seen as a limitation. Usually the vial is cleaned, before entering in cabinet, could stay for a long time within the cabinet, together with other drugs, and this could be potentially an environment that affects the microbiological contamination of the vials.
2. Also, the vial that is opened in the cabinet and then stored, suffer a instability in temperature. The drug that is stored from 2-8º were in cabinet for a while, and then is stored again. This could be pottentially a source for degradation that should be discussed. What temperature were the vials exposed before re-storing?
3. Figure 3 is quite hard to read. Authors state that line are superimposing; the way is presented seems that there are differences between them. Could you add some more explanation or reproduce it in a different plot.
Author Response
"Please see the attachment."

Round 2
Reviewer 1 Report
Comments and Suggestions for Authors
After having reviewed the revised manuscript, I believe the authors have improved it sufficiently to allow it to be accepted.